# The Effectiveness of Additional Core Stability Exercises in Improving Dynamic Sitting Balance, Gait and Functional Rehabilitation for Subacute Stroke Patients (CORE-Trial): Study Protocol for a Randomized Controlled Trial

**DOI:** 10.3390/ijerph18126615

**Published:** 2021-06-19

**Authors:** Rosa Cabanas-Valdés, Lídia Boix-Sala, Montserrat Grau-Pellicer, Juan Antonio Guzmán-Bernal, Fernanda Maria Caballero-Gómez, Gerard Urrútia

**Affiliations:** 1Physiotherapy Department, Faculty of Medicine and Health Science, Universitat Internacional de Catalunya, Sant Cugat del Valles, 08195 Barcelona, Spain; 2Rehabilitation Unit, Fundació Hospital de la Santa Creu de Vic, 08500 Vic Barcelona, Spain; lboix@hsc.chv.cat; 3Physiotherapy Department, University of Vic-Central University of Catalonia (UVIC-UCC), 08500 Vic Barcelona, Spain; 4Rehabilitation Unit, Hospital-Consorci Sanitari de Terrassa, 08221 Barcelona, Spain; MGrauP@cst.cat; 5Physiotherapy Department, Autonomous University of Barcelona, 08193 Barcelona, Spain; 6Rehabilitation Unit, Hospital Sagrat Cor Germanes Hospitalaries, Martorell, 08760 Barcelona, Spain; jguzman.hsagratcor@hospitalarias.es; 7Physical Medicine and Rehabilitation Sabadell, Consorci Corporació Sanitària Parc Taulí, 08208 Barcelona, Spain; FcaballeroG@tauli.cat; 8Institut d’Investigació Biomèdica Sant Pau (IIB Sant Pau), 08041 Barcelona, Spain; GUrrutia@santpau.cat; 9CIBERESP, 28029 Madrid, Spain

**Keywords:** stroke, exercise therapy, sitting position, postural balance, gait, core stability training, trunk exercises

## Abstract

Background: Trunk impairment produces disorders of motor control, balance and gait. Core stability exercises (CSE) are a good strategy to improve local strength of trunk, balance and gait. Methods and analysis: This is a single-blind multicenter randomized controlled trial. Two parallel groups are compared, and both perform the same type of therapy. A control group (CG) (*n* = 110) performs conventional physiotherapy (CP) (1 h per session) focused on improving balance. An experimental group (EG) (*n* = 110) performs CSE (30 min) in addition to CP (30 min) (1 h/session in total). EG is divided in two subgroups, in which only half of patients (*n* = 55) perform CSE plus transcutaneous electrical nerve stimulation (TENS). Primary outcome measures are dynamic sitting, assessed by a Spanish version of Trunk Impairment Scale and stepping, assessed by Brunel Balance Assessment. Secondary outcomes are postural control, assessed by Postural Assessment Scale for Stroke patients; standing balance and risk of fall assessed by Berg Balance Scale; gait speed by BTS G-Walk (accelerometer); rate of falls, lower-limb spasticity by Modified Ashworth Scale; activities of daily living by Barthel Index; and quality of life by EQ-5D-5L. These are evaluated at baseline (T0), at three weeks (T1), at five weeks (end of the intervention) (T2), at 17 weeks (T3) and at 29 weeks (T4). Study duration per patient is 29 weeks (a five-week intervention, followed by a 24-week post-intervention).

## 1. Introduction

Strokes have a high morbidity, and also result in up to 50% of survivors being chronically disabled. Thus, a stroke is a disease of public health importance with serious economic and social consequences [1]. Over 80% experience a balance disturbance in the subacute phase [2]. It reduces ability to perform daily tasks, and at six months after a stroke, 40% of stroke survivors have difficulties with basic activities of daily living (ADL), and 30% report participation restrictions, even at four years after a stroke [3].

This balance dysfunction is usually due to a combination of reduced limb, pelvis and trunk motor control, altered sensation (proprioception) of one side, and sometimes centrally determined alteration in body representation [4]. Trunk impairment is closely associated with postural imbalance and functional performance instability in gait [5] and in standing balance, increasing the risk of falls and fear of falling [6]. It can lead to reduction in independence to undertake ADL and reduced quality of life (QoL) [7].

Falls may have serious physical and psychological consequences, including increased risk of hip fracture (usually on the weaker side) and greater mortality and morbidity compared to people who have not had a stroke [8]. Fear of falling may lead to decreased physical activity, social isolation and loss of independence [9]. Any postural control disorder increases the risk of falling and injury [10]. Consequently, balance improvement is associated with decreased risk and fear of falling, as well as with improved QoL [11]. 

Balance is achieved through an interaction of central anticipatory and reflexive actions assisted by the active and passive restraints caused by the muscular system. The perception of verticality is based on the construction of a body-centred frame of reference in the gravitational environment. This frame of reference consists of afferent visual, vestibular information and somatosensory information of the body [12], which translate into an adequate motor response in order to guarantee both the anticipatory and adaptive aspects of balance control. In particular, the position sense of the trunk could be important to provide information about the alignment of the trunk in relation to gravity [13]. Trunk control and dynamic sitting balance are a fundamental requirement to be able to lead an independent life and to carry out ADL, such as combing, dressing or going to the bathroom. When walking, the human body is never balanced; most of the time the trunk is supported by one leg and the centre of mass “falls” onto the contralateral side. Therefore, trunk control in gait is an essential component [5].

Sitting lateral balance control appears to be the function critically affected by strokes, as well as the most sensitive to functional changes induced by rehabilitation [14]. Therefore, this will be the first goal to achieve with the neurorehabilitation treatment. Another aim is to achieve walking ability at discharge for minimising activity limitations and for maximising QoL. The control of the lower trunk, pelvis and leg muscles allows maintaining the centre of mass inside a stable base of support [15].

Rehabilitation is offered to all stroke survivors in the subacute phase after receiving initial medical treatment, in order to reduce their disability and accelerate their independence and resume ADL. An earlier and more intensive rehabilitation program in the early phase of a stroke is related to the good recovery of walking and functional independence status according to the concept “time is brain recovery” [16]. Furthermore, it is important to conduct rehabilitation trials during the initial days and weeks after a stroke, since it is then when spontaneous biological recovery takes place and when rehabilitation is delivered in the “real world” [17]. The first week until the first month post-stroke (acute and early subacute phase) is critical for neuroplasticity [18,19]. Recovery after a stroke follows a curve; it is not linear, with most of the improvement occurring during the first few days to six months [20]. 

A Cochrane systematic review concludes that 30 to 60 min per day delivered five to seven days per week is effective to recover function and mobility after stroke [21]. A good rehabilitation strategy, which might help improve trunk performance, trunk control and dynamic sitting balance [22], is approaches using trunk training therapeutic exercises [23] (today commonly known as Core Stability Exercises, or CSEs). Recent studies suggest that core strengthening plays a critical role in maintaining balance [24,25,26,27], functional mobility [28], gait, fear of falls and in improving anticipatory postural adjustment [29] in stroke survivors. Findings suggest that CSEs plus conventional physiotherapy has a positive long-term effect on improving dynamic sitting, standing balance and gait at three months after the end of treatment [30]. However, there is no consensus about which are the most effective intervention parameters, about intensity, and how early training exercises in the stroke subacute phase should be [31].

CSEs are voluntary movements that aim at promoting the neuromuscular control, coordination, strength and endurance of muscles that are central to maintaining dynamic stability of the spine and trunk. It is the ability to control the position and motion of the trunk over the pelvis and leg that allows optimal production, transfer and control of force and motion to the terminal segment in integrated kinetic chain activities [32,33]. It is essential to providing a solid base of core to exert or resist force, as it stabilizes the pelvis and spinal column for “proximal stability for distal mobility” [34]. Static core functionality is the ability of the core to align the skeleton to resist a force that does not change. The body core corresponds to the synergy 2, described by Israely et al. [35]. 

Several studies have shown that transcutaneous electrical nerve stimulation (TENS) applied to the trunk muscles during CSEs training could increase the motor output of trunk muscles. CSEs training combined with TENS could be more effective than CSEs alone for improving dynamic sitting balance [36]. TENS has shown to excite large sensory fibers, predominantly in the A-beta range through the cutaneous stimulation of muscles—it increases the excitability of the sensorimotor cortex [37].

Any intervention in the stroke subacute phase that reduces disability will probably be cost-effective. It should be noted that stroke rates are multiplied by 10 in the population over 75 years of age; this population usually suffers from sarcopenia, in addition to muscle weakness caused by strokes. For this reason, it is important to activate and strengthen the core muscles, since it has been shown that muscle atrophy and a significant impairment of postural reactive responses in the trunk rapidly occur [38]. This has led to shorter inpatient stays at the hospital becoming increasingly essential to have rehabilitation programs that are more efficient, which implies that patients have greater autonomy when discharged from hospital. 

There are few high-quality large multicenter randomized controlled trials (RCT) with patients recruited within 30 days after a stroke [39]. There is a clear need for larger trials conducted early after stroke in real-world clinical settings [40]. To determine the effectiveness, safety and optimal training parameters of CSEs, homogeneous post-stroke populations and follow-up measures are necessary [41]. 

The primary objective of this study is to evaluate the effectiveness of CSEs protocol (with and without TENS), in addition to conventional physiotherapy (CP) to improve dynamic sitting balance and gait (stepping) at short/mid-term in the subacute phase of a stroke. Secondary objectives are to evaluate the effectiveness of CSEs (with and without TENS) in addition to CP to improve postural control, standing balance, fall rate, risk of falls, gait speed, lower limb spasticity, ADL and QoL. Another secondary objective is to explore the sustainability of the effects of CSEs over time. It is important to know whether the treatment effects are sustainable over time, or if continuous therapeutic input is necessary to maintain the level of function even after being discharged (home).

## 2. Materials and Methods

### 2.1. Study Design and Setting

This study is an assessor-blinded, multicenter RCT, with a five-week treatment period followed by a three-to-six-month follow-up. It follows the consensus-based core recommendations from the stroke recovery and rehabilitation expert group [42] and the SPIRIT statement [43]. Participants will be randomly allocated (at a ratio of 2:1:1) to the control group (CG) (*n* = 110) or the experimental group (EG with and without TENS) (*n* = 110), <15 and >15 days after stroke. Patients are being recruited by inpatient rehabilitation hospitals in six centers in Catalonia: Hospital Universitari Parc Taulí de Sabadell; Hospital-Consorci Sanitari de Terrassa; Fundació Hospital de la Santa Creu de Vic; Hospital Sagrat Cor Germanes Hospitalaries de Martorell; Hospital Sociosanitari Mutuam Girona; and Centre Fòrum Parc de Salut Barcelona.

### 2.2. Recruitment

Treatments are randomly assigned using a computer program. To guarantee allocation concealment, treatments are assigned centrally via the web through Clinapsis^®^, an application designed to assist in the design and management of epidemiological and non-commercial clinical studies Clinapsis® Patients are recruited and screened for eligibility in three consecutive steps. Firstly, the principal investigator of each hospital is thoroughly briefed concerning the inclusion and exclusion criteria of the study, since they provide therapists with the information for possible inclusion. Secondly, the main researcher gives information about the study to potential participants, including the objective and description of the study, the duration, and risks and benefits. If the patients are interested in the study, an appointment is made to provide more detailed information and to answer questions. When the patient agrees to participate in the study, the informed consent is signed before obtaining the medical record, in order to guarantee privacy. Lastly, after obtaining informed consent, the patients are screened by the primary investigator to assure inclusion. 

After group allocation and before starting treatment (T0), pre-intervention tests are performed to assess the baseline values of primary and secondary outcome measures. At week 3, an assessment (T1) of only primary outcomes is performed. Within one day after completing the intervention (T2), data will be collected for all efficacy outcomes (see outcome measures section). The same data will be collected at three months after the end of the intervention (T3), and again at six months (T4). During the five-week intervention period, each session data will be collected regarding intervention adherence (number of sessions and duration), physiotherapy intensity, which exercises were performed for each patient and their incidences. All visits and efficacy assessments are performed at the rehabilitation center where patients have been initially treated for five weeks (whenever possible during routine clinical follow-up visits). Only when the patient is not able to personally attend the site due to a medical condition, the assessment takes place at home.

All data are recorded on-line using an electronic data form Clinapsis^®^, available from the study coordinating center. All investigators were trained in the use of the application and have a help guide, as well as a consultation service directly with the logistics coordinator of the study. Access to the study database is restricted to authorised study personnel by password.

For the calculation of the sample size, we have assumed that conventional rehabilitation will be associated with a clinically relevant change in the Trunk Impairment Scale at five weeks compared to the baseline. The minimal clinically relevant difference has been established as 3-point [44]. We have also assumed that rehabilitation by CSEs program will add a benefit of 1.6 points at five weeks, equivalent to 10% in the scale. That is, the experimental group with CSE will present a change of 4.6 points at five weeks with respect to the baseline situation (intragroup). Assuming a common standard deviation of 4 [45], and estimating a 10% lost at follow-up, accepting an alpha risk of 0.05 and a beta risk of less than 0.2 in a bilateral contrast, it will be necessary to include 110 patients in each group to detect a difference between groups of 1.6 points or higher on the total S-TIS 2.0 scale. The calculation of the sample size was done with the GRANMO program.

If a patient, either in CG or EG, has to withdraw from the treatment sessions due to a transient disease or mind trauma, he or she may be re-included if the dropout period is shorter than 10 days. Patients are allowed to withdraw from the study for any reason, and no adverse events have been described previously.

#### Patient and Public Involvement Subsection

Patients and the public were not involved in any way in the co-production of this research.

### 2.3. Blinding

Due to the nature of the interventions, the study has a single-blind design. Therapists and participants cannot be blinded to treatment allocation. To avoid detection bias, efficacy outcomes are evaluated by an independent assessor blinded to the intervention. Each center has a therapist evaluator. They had a training day by principal investigator for the correct use of scales and questionnaires. This information is available online and in the paper. Furthermore, statistical analysis will be conducted, blinded to the allocation.

### 2.4. Selection Criteria 

Patients will have to meet the following eligibility criteria to be included in the study.

Inclusion criteria

First ever-stroke ≤ 30 days (diagnostic criteria according to the World Health Organisation definition; corresponding to ICD-9 code 434) whether cortical or subcortical, and ischemic or hemorrhagic.Unilateral localisation of the stroke verified by computed tomography; if a patient shows previous problems, but does not have any neurological or clinical impairment, he/she would be included in the study.Both sexes and age ≥ 18 years old.Ability to understand and execute simple instructions.Impairment of sitting balance assessed by the Spanish Version of Trunk Impairment Scale.2.0 (S-TIS 2.0) ≤10 points [46,47].Severity of stroke by the Spanish National institute of Health Stroke Scale (S-NIHSS) [48] score ≥ 2 points.

Exclusion criteria

Modified Rankin Scale [49] > 2 points before stroke.Concurrent neurological disorder (e.g., Parkinson’s disease) or major orthopedic problem (e.g., amputation) that hampers sitting balance.Relevant psychiatric disorders that may prevent from following instructions.Other treatments that could influence the effects of the interventions.Contraindication to physical activity (e.g., heart failure).Use of cardiac pacemakers.Patients with hemorrhagic strokes that have undergone surgery for intracranial decompression.Patients whose stroke occurs exclusively and only in the cerebellum and brainstem. Patients whose main stroke is localised on another area and who also have a small lesion in the cerebellum and brainstem would not be excluded.

### 2.5. Interventions

The interventions are performed five days a week for five weeks. EG participants receive 12.5 h (30 min per day) of additional core stability exercises (with or without TENS). All individuals will perform conventional physiotherapy (EG:30 min and CG: 1 h). All interventions will be performed by trained experts in neurological physiotherapy with extensive/over five years of experience treating stroke survivors and with a Master’s Degree in Neurology. Before starting the study, a one-day training session was carried out in order to standardise the procedures and provide the physiotherapists with specific training in the CSEs program and TENS by the clinical director. If there are doubts, they can always get in touch with the clinical coordinator and principal investigator. Each centre has a dossier with the program and videotapes of all CSEs and their explanations. In the meeting with all therapists of the different hospitals involved, a protocol for conventional physiotherapy focusing on improving patients’ disabilities was agreed upon.

The follow-up period is not controlled. At this stage, patients will not follow a specific treatment supervised by the research team, i.e., “usual care”, and the duration can also be variable. Patients of experimental and control group may continue to perform CP and/or aerobic-based therapy as prescribed by the responsible physician, or on their own initiative (private physiotherapy) if they wish. In this case, this additional physiotherapy will not be provided in the same rehabilitation unit by the same previous physiotherapist, but in outpatient physiotherapy centers. Conventional therapy and aerobic-based therapy for long term are recorded.

#### Intervention Description

CSEs were designed to improve the endurance of core muscles that stabilise the trunk and pelvis. The CSEs program consists of 23 exercises focused on trunk muscle strengthening, proprioception, selective movements of the trunk and pelvis muscle and coordination. They are carried out in the supine position, sitting on a stable surface and on an unstable surface (physioball). The exercises involve changes in the position of the body with or without resistance. Training is determined by the patient’s ability to perform easy exercises and their progress to more challenging exercises. Adequate rest periods are allowed between exercises. Start with simple movements and progress to multi-plane movements when the basics are secure. For each exercise, ten repetitions are performed, two of them with the eyes closed. If the patient is afraid in sitting exercises on a physioball, they should not be forced to close them. It is not necessary that the patient perform all exercises per session. It may be performed depending on the patient’s possibilities, but most important is that they were performed 30 min of CSEs, and were recorded. For monitoring the perceived individual’s exertion of CSE training, the Borg scale of perceived exertion will be used [50]. The intensity of the effort perceived by the patient during training will remain moderate (4–5 points-score). They would not move on to higher levels until they had mastered the exercise they were engaged in. The physiotherapists perform the therapy with their hands on the patient to ensure proper quality of movement, and do not participate in the patient’s evaluation. When the patient performs them correctly, they will perform them again alone (see Figure 1).

Transcutaneous electrical nerve stimulation (TENS). The high frequency of TENS is 100 Hz; 0.2 µs pulse width, mm-diameter electrodes placed on the skin over the lumbar erector spinae muscles (3 cm lateral to the L3 and L5 spinous process). The intensity of stimulation is twice the sensory threshold (the minimum intensity the subject could feel), which was barely below the motor threshold. The pulse trains were delivered with a two-channel stimulation device (Cefar PRIMO PRO, tens. 2 channels).

The comparator of EG is CP. CP involves different interventions improving functional capacity and reducing disability. The common feature of CP is that it consists of a treatment performed by the physiotherapist according to the degree of affectation of the particular patient, and according to the degree of accomplishment of the objectives set. CP may consist in a variety (or combination) of multiple components such as tone normalisation based on hands-on therapy interventions with sensory feedback by manual contact [51], passive or active joint mobilisation for maintaining range of motion, and active or active-assisted exercise of affected side [52], as well as sit to stand training [53] with or without gripping on wall bars, sitting balance (without core stability exercises), standing balance training [54] and gait re-education (walking between parallel bars or with a physiotherapist). 

Co-interventions: during the five-week intervention phase, patients can receive other usual types of rehabilitation management (such as occupational therapy, speech therapy and neuropsychology) in accordance with local practices. All these co-interventions are being recorded, and measures will be taken to control for possible performance bias. Interventions normally last 30–45 min. In this phase of the stroke, the patient is usually highly motivated to recover and especially regain balance in sitting, standing and walking, and thus be able to perform their ADLs. It is very unlikely that the treatment cannot be carried out. This could only be the case for patients with a cognitive impairment, but these are initially excluded from the study. 

### 2.6. Participant Timeline

The study has five assessments: T0 (baseline), T1 (week three, only primary outcomes), T2 (week five, end-point, 25 sessions), T3 (week 17) and T4 (week 29). Study duration per patient: 29 weeks (See Figure 2).

### 2.7. Outcomes Measures

Primary outcome measures include: Dynamic sitting balance and coordination measured by S-TIS 2.0 [48]. This scale is a Spanish version of the Trunk Impairment Scale version 2.0 [55]. This scale aims to evaluate the trunk in patients who have suffered a stroke. The dynamic subscale contains items on the lateral flexion of the trunk and unilateral lifting of the hip. To assess the coordination of the trunk, the individual is asked to rotate the upper or lower part of his or her trunk six times, initiating the movements either from the shoulder girdle or from the pelvic girdle, respectively. There are two subscales; the first one has 10 items and the second one has six. The highest possible total score is consequently 16 points, which indicates an optimal dynamic sitting balance and sitting coordination. If the patient cannot maintain a sitting position for 10 s without back and arm support, with hands on thighs, feet in contact with the ground and knees bent at 90° (starting position), the total score for the scale is 0 points. This scale is utilised for inclusion criteria, and at T0, T1, T2, T3 and T4.Gait by stepping section of Brunel Balance Assessment (BBA) [56]. It is designed to assess functional balance for people with a wide range of abilities, and has been tested specifically for use post-stroke. There are three sections to the assessment: sitting, standing and stepping. In this study, only the stepping section is utilised. It consists of six levels to assess standing functional balance and a 5-m walk. At each level, the patient receives a score for his/her efforts. This gives an indication on whether the patient is improving within a level, even if he/she is not able to progress to the next level. The score also reflects how well the individual is functioning within that stepping section. The higher score is six points, and the individual is able to walk 5 m independently. Stepping is evaluated at T0, T1, T2, T3 and T4.

Secondary outcome measures include: Sitting functional balance is assessed by the Spanish version of Function in Sitting test (S-FIST) [57]. It is a bedside evaluation of sitting balance and functional sitting everyday activities that assess sensory, motor, proactive, reactive and steady balance factors. The S-FIST consists of 14 tested parameters with an ordinal scale (0–4) for each test item, with 0 indicating the lowest level of function and 4 the highest level. Each participant sat at the edge of a standard hospital bed without air mattresses, with the proximal thigh (1/2 femur length) supported by the bed. The bed height was adjusted and a step stool was used if necessary to bring the hips and knees to approximately 90° flexion, with both feet flat on the floor or stool. The higher score is 56 points. Sitting functional balance is evaluated at T0, T2, T3 and T4.Standing balance and risk of falling is evaluated by Berg Balance Scale (BBS) [58,59]. It provides a psychometrically sound measure of balance impairment. It is used objectively determine a patient’s ability (or inability) to safely balance during a series of predetermined tasks. It is a 14-item scale; patients must maintain positions and complete moving tasks of varying difficulty. In most items, patients must maintain a given position for a specified time. Each item consists of a 5-point ordinal scale ranging from 0 to 4, with 0 indicating the lowest level of function and 4 the highest. A score of 56 indicates functional balance. A score of < 45 indicates that individuals may be at greater risk of falling. BBS is assessed at T0, T2, T3 and T4.Postural control is evaluated by the Spanish version of Postural Assessment Scale for Stroke (S-PASS) [60]. It was designed specifically for patients with a stroke, regardless of postural competence. It has two subscales: mobility and balance. The first measures the patient’s ability to change position from lying, sitting and standing, and the second in maintaining stable postures in sitting and standing. The S-PASS consists of 12 items with a 4-point scale, where items are scored from 0–3. The higher score is 36 points, indicating an optimal postural control. It is evaluated at T0, T2, T3 and T4.Lower limb spasticity by Modified Ashworth Scale (MAS) [61]. This tool measures resistance during passive soft-tissue stretching of muscle. It is performed while the assessor moves the hip adductors, knee extensors and ankle plantar flexors in the supine and lateral position. The MAS is assessed at T0, T2, T3 and T4.ADL by Barthel Index (BI) [62]. This shows the degree of independence of a patient from any assistance. It covers 10 domains of function (activities): bowel and bladder control, as well as help with grooming, toilet use, feeding, transfers, walking, dressing, climbing stairs and bathing. The ADL is evaluated at T0, T2, T3 and T4.Health-related quality of life is measured by the Spanish-version of 5-Dimensions Questionnaire (EQ-5D-5L) [63,64]. It is a generic patient’s health-related quality of life measurement with evidence of good reliability and validity in various disease populations, including strokes. Patients chose five levels of severity (1, no problem; 2, slight problem; 3, moderate problem; 4, severe problem; and 5, unable to function/extreme problem) in five dimensions (mobility, self-care, usual activity, pain/discomfort and depression/anxiety), and rated their overall health status via the EQ-VAS. Quality of life is assessed at T0, T2, T3 and T4.Rate of falls is measured by a specific registry created specifically for this study. The outcome is defined as the average number of falls per patient during the intervention period and follow-up. It is recorded at T0 (falls before stroke), T2, T3 and T4.Gait speed is assessed by BTS G-Walk. It is a wireless system consisting of an inertial sensor composed by a triaxial accelerometer, a magnetic sensor and a triaxial gyroscope that was positioned on S1 vertebrae. From the data acquired, the system extrapolates all spatial-temporal gait. The patient walks for one minute without being aided; this variable is only performed if the patient has a 6-point stepping section of BBA.

#### Baseline Assessment

Information concerning stroke diagnosis, medical history and stroke onset will be acquired from patient records, from which participant characteristics will also be collected: sex, age, medication use, co-morbidities, side and location of the lesion, days post-stroke, stroke severity as assessed by NIHSS and modified Rankin Scale (mRS) and immediate treatment for stroke (thrombolysis/thrombectomy).

### 2.8. Statistical Analysis

The main analysis population will be defined by intention-to-treat, comprising all randomised participants with a baseline assessment, regardless of later events such as protocol violations, missing data or loss to follow-up. Missing data will be imputed and the impacts of imputation on results will be explored. Secondary analyses will be conducted, restricted to the population of participants who followed the study protocol and had no missing data.

General linear models with repeated measures design will be used to test the effect of the Core Stability program (with or without TENS) on the change in the outcome of interest (total S-TIS 2.0 score or any other outcome) between the baseline and follow up (five weeks), using a repeated measures design with two levels (baseline, five weeks), as well as a two-level factor testing the intervention (usual care). The models will adjust for clinically relevant covariates, such as site, age, baseline levels of the dependent outcomes or severity of stroke. Similar models will be built to explore the effect of Core Stability at different time points (three months; six months). Similar models will test the effect of TENS added to Core Stability on primary and secondary outcomes, as well as primary and secondary endpoints.

## 3. Discussion

Motor rehabilitation after stroke continues to be an area in need of substantial financial and scientific investment. There is also a need for more pragmatic trials to test interventions in a way that assists their translation into clinical practice. The analysis of recovery in subacute phase profiles is important, as this information can provide a more specific plan for stroke rehabilitation in this phase. Exercise intensity and type is not only the most challenging parameter to determine, but it is also the most critical one to ensure that a dose is safe, attainable and adequate to elicit a training effect. A previous study demonstrated that the repetitive movement may be the best to stimulate cerebral neural plasticity [65]. CSEs are repetitive movements in different positions. CSEs in this study are not only performed in the supine position – when possible, they are performed in the sitting position. Training of vertical trunk function is important because humans stand and move using two legs [66].

We hope that dynamic sitting balance and gait will be better in EG than CG. Concerning gait, a minimal detectable change of 1 point for BBA (stepping) would be a good result [67]. Regarding standing balance, a difference change of ± 6 points for BBS is necessary to be 90% confident about a genuine change during inpatient stroke rehabilitation [68,69]. It has been reported more recently that a change of 4.8 points in the FIST is clinically important [70]. 

The co-activation of the diaphragm, transversus abdominis and internal oblique improves postural control [71] – these muscles are part of a core. We assess postural control by S-PASS. A minimally detectable change of 2.22 points on S-PASS could be a good result [72]. It is important to consider measures of activity and functional outcomes when determining whether an intervention is effective. For this reason, we chose the Barthel Index and EQ-5D-5L. There is little evidence that core stabilisation improves ADL [73]. A minimally clinically important difference on Barthel Index is ≈2 points [74]. As for EQ-5D-5L, values of 0.10 on the EQ-Index and 8.61–10.82 on the EQ-VAS are likely to have a clinically important change [75]. Assessing participation outcomes such as quality of life is necessary, according to Gamble et al. [76]. 

A decrease in gait velocity and double support time could represent an attempt at increasing postural stability to reduce fall risk [69]. Specifically, decreased velocity may reduce the body’s momentum, increasing the likelihood of recovering from a loss of balance. A walking speed of 0.8 m/s is predictive of weak functional abilities, while a speed of 0.6 m/s establishes a threshold below which the risk of falling is critical [77]—the minimally clinically important difference in stroke patients (0.6; 0.8 m/s) [78]. 

One of the first objectives of rehabilitation in the subacute phase of the stroke is to achieve optimal trunk control and dynamic balance while sitting. The proposal of this study is to achieve this by training the core muscles. This results in a better balance in sitting, standing, and in a more efficient gait [79]. The earlier these patients can be autonomous, the less time they will be inactive [80], and in this way the adverse effects of immobilisation can be reduced [81]. Additional trunk rehabilitation is beneficial in improving gait performance in sub-acute stroke [82]. Therefore, if the results of this study are positive, we recommend to incorporate these exercises as early as possible in a stroke rehabilitation program. 

Another important goal of this study is to discover whether the treatment effects are sustainable over time, or if the continuous therapeutic input is necessary to maintain the level of function even after the discharge home [83]. In our knowledge, there is no systematic review about the long-term effects of core stability exercises on improving balance and gait in a stroke subacute phase.

## Figures and Tables

**Figure 1 ijerph-18-06615-f001:**
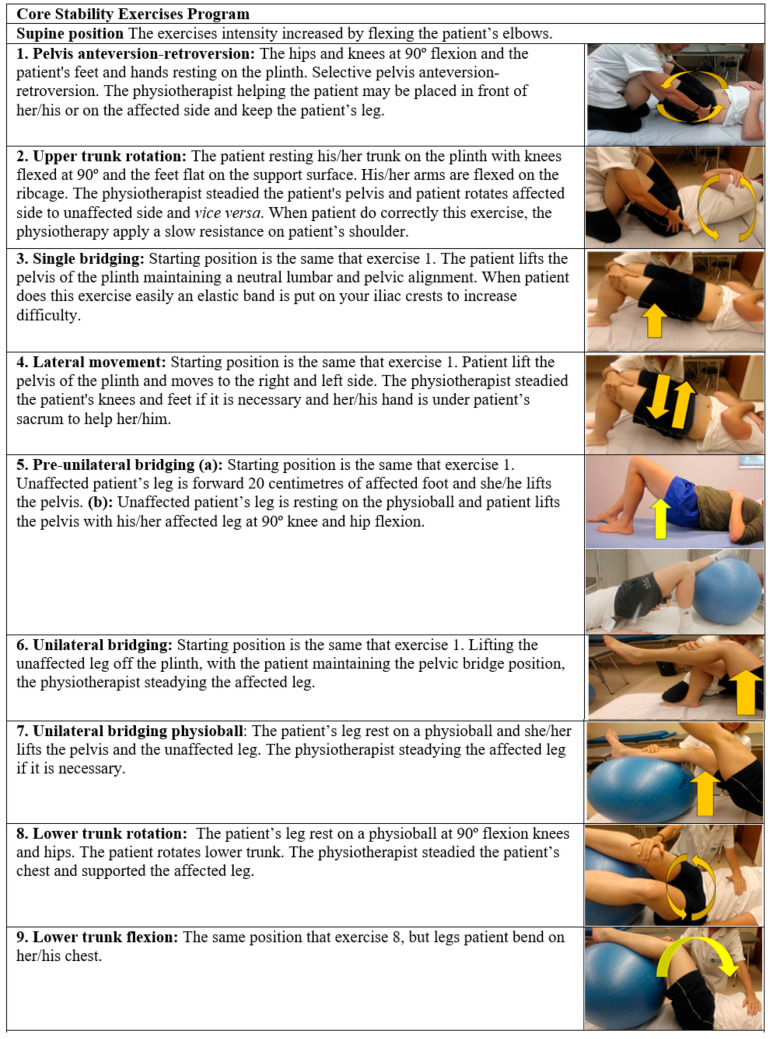
Core stability exercises intervention.

**Figure 2 ijerph-18-06615-f002:**
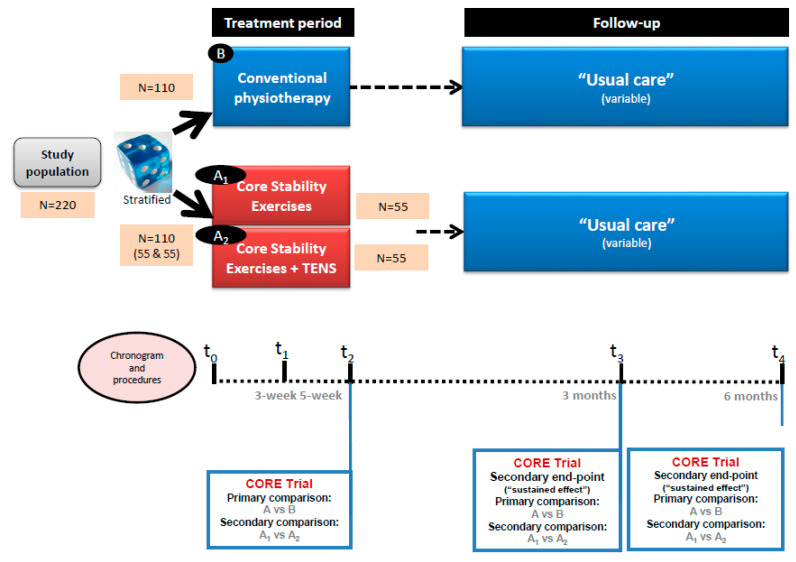
Study scheme.

## Data Availability

Not applicable.

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
