# Peer review of "The Effectiveness of Additional Core Stability Exercises in Improving Dynamic Sitting Balance, Gait and Functional Rehabilitation for Subacute Stroke Patients (CORE-Trial): Study Protocol for a Randomized Controlled Trial"

_ijerph, 2021, doi:10.3390/ijerph18126615_

Round 1
Reviewer 1 Report
The manuscript is on a study protocol of a randomized, parallel, clinical trial comparing the effects of two training interventions. Overall the article is well written and it was easy to follow. Even though, is missing clear information on the measurements (description of the different instruments and their application).
Also regarding interventions is not clear how many days per week the patients are going to perform. It is not clear to me if the number the adequacy of the number of exercises. The patients are going to perform 22 exercises per session?
Author Response
Thanks for your suggestions.
I am responding your questions:
Line 241 page 6 it was added "The interventions are performed 5 days a week per 5 weeks.EG participants are receiving 12,5 hours (30 minutes per day) of additional core stability exercises (with or without TENS). All individuals will perform conventional physiotherapy (EG:30 minutes and CG: 1 hour)".
The patients are going to perform all exercises per session? No, it is not necessary. It was added line 271 "For each exercise, ten repetitions are performed, two of them, with the eyes closed. If the patient is afraid in sitting exercises on a physioball, they should not be forced to close them. It is not necessary that the patient perform all exercises per session. It may be performed depending of the patient possibilities, the most important is that they were performed 30 minutes of CSEs and it are recorded".
The description of the different instruments and their application were added. It is in page 12 and 13 in yellow colour.
Reviewer 2 Report
In general, the manuscript is well written except for the minor language corrections. The topic is very interesting for coaches and sports scientists to improve the rehabilitation training concepts. In addition, the novelty in this manuscript was known whether the treatment effects are sustainable over time or if the continuous therapeutic input is necessary to maintain the level of function even after discharge home.
The important comment is, the discussion missed the variety of references that provide more evidence in this part. It must be modified.
Author Response
Thank you for your suggestions.
The discussion section was modified with new references.
Halder, P.; Sterr, A.; Brem, S.; Bucher, K.; Kollias, S.; Brandeis, D. Electrophysiological evidence for cortical plasticity with movement repetition. Eur. J. Neurosci. 2005, 21, 2271–2277, doi:10.1111/j.1460-9568.2005.04045.x.
Kinoshita, K.; Ishida, K.; Hashimoto, M.; Nakao, H.; Shibanuma, N.; Kurosaka, M.; Otsuki, S. A vertical load applied towards the trunk unilaterally increases the bilateral abdominal muscle activities. J. Phys. Ther. Sci. 2019, 31, 273–276, doi:10.1589/jpts.31.273.
Yoon, H.S.; Cha, Y.J.; You, J.H. Effects of dynamic core-postural chain stabilization on diaphragm movement, abdominal muscle thickness, and postural control in patients with subacute stroke: A randomized control trial. NeuroRehabilitation 2020, 46, 381–389, doi:10.3233/NRE-192983.
Yoon, H.S.; Cha, Y.J.; You, J.S.H. The effects of dynamic core-postural chain stabilization on respiratory function, fatigue and activities of daily living in subacute stroke patients: A randomized control trial. NeuroRehabilitation 2020, 47, 471–477, doi:10.3233/NRE-203231.
Gamble, K.; Chiu, A.; Peiris, C. Core Stability Exercises in Addition to Usual Care Physiotherapy Improve Stability and Balance After Stroke: A Systematic Review and Meta-analysis. Arch. Phys. Med. Rehabil. 2021, 102, 762–775.
Karthikbabu, S.; Chakrapani, M.; Ganesan, S.; Ellajosyula, R.; Solomon, J.M. Efficacy of Trunk Regimes on Balance, Mobility, Physical Function, and Community Reintegration in Chronic Stroke: A Parallel-Group Randomized Trial. J. Stroke Cerebrovasc. Dis. 2018, 27, 1003–1011,
Chen, X.; Gan, Z.; Tian, W.; Lv, Y. Effects of rehabilitation training of core muscle stability on stroke patients with hemiplegia. Pakistan J. Med. Sci. 2020, 36, 461–466, doi:10.12669/pjms.36.3.1466.
van Criekinge, T.; Hallemans, A.; Herssens, N.; Lafosse, C.; Claes, D.; de Hertogh, W.; Truijen, S.; Saeys, W. SWEAT2 Study: Effectiveness of Trunk Training on Gait and Trunk Kinematics after Stroke: A Randomized Controlled Trial. Phys. Ther. 2020, 100, 1568–1581, doi:10.1093/ptj/pzaa110.
Karthikbabu, S.; Verheyden, G. Relationship between trunk control, core muscle strength and balance confidence in community-dwelling patients with chronic stroke. Top. Stroke Rehabil. 2021, 28, 88–95, doi:10.1080/10749357.2020.1783896.